# A Systematic Study on the Degradation Products Generated from Artificially Aged Microplastics

**DOI:** 10.3390/polym13121997

**Published:** 2021-06-18

**Authors:** Greta Biale, Jacopo La Nasa, Marco Mattonai, Andrea Corti, Virginia Vinciguerra, Valter Castelvetro, Francesca Modugno

**Affiliations:** 1Department of Chemistry and Industrial Chemistry, University of Pisa, 56124 Pisa, Italy; greta.biale@gmail.com (G.B.); marco.mattonai@dcci.unipi.it (M.M.); andrea.corti@unipi.it (A.C.); virgi-vinci@hotmail.it (V.V.); valter.castelvetro@unipi.it (V.C.); francesca.modugno@unipi.it (F.M.); 2National Interuniversity Consortium of Materials Science and Technology, 50121 Florence, Italy; 3CISUP—Center for the Integration of Scientific Instruments of the University of Pisa, University of Pisa, 56124 Pisa, Italy

**Keywords:** microplastics, polymer degradation, artificial ageing, polyolefins, polystyrene, polyethylene terephthalate

## Abstract

Most of the analytical studies focused on microplastics (MPs) are based on the detection and identification of the polymers constituting the particles. On the other hand, plastic debris in the environment undergoes chemical and physical degradation processes leading not only to mechanical but also to molecular fragmentation quickly resulting in the formation of leachable, soluble and/or volatile degradation products that are released in the environment. We performed the analysis of reference MPs–polymer micropowders obtained by grinding a set of five polymer types down to final size in the 857–509 μm range, namely high- and low-density polyethylene, polystyrene (PS), polypropylene (PP), and polyethylene terephthalate (PET). The reference MPs were artificially aged in a solar-box to investigate their degradation processes by characterizing the aged (photo-oxidized) MPs and their low molecular weight and/or highly oxidized fraction. For this purpose, the artificially aged MPs were subjected to extraction in polar organic solvents, targeting selective recovery of the low molecular weight fractions generated during the artificial aging. Analysis of the extractable fractions and of the residues was carried out by a multi-technique approach combining evolved gas analysis–mass spectrometry (EGA–MS), pyrolysis–gas chromatography–mass spectrometry (Py–GC–MS), and size exclusion chromatography (SEC). The results provided information on the degradation products formed during accelerated aging. Up to 18 wt% of extractable, low molecular weight fraction was recovered from the photo-aged MPs, depending on the polymer type. The photo-degradation products of polyolefins (PE and PP) included a wide range of long chain alcohols, aldehydes, ketones, carboxylic acids, and hydroxy acids, as detected in the soluble fractions of aged samples. SEC analyses also showed a marked decrease in the average molecular weight of PP polymer chains, whereas cross-linking was observed in the case of PS. The most abundant low molecular weight photo-degradation products of PS were benzoic acid and 1,4-benzenedicarboxylic acid, while PET had the highest stability towards aging, as indicated by the modest generation of low molecular weight species.

## 1. Introduction

Due to the steadily increasing production of plastic materials since the second half of the 20th century, the mismanagement of a significant fraction of plastic waste has resulted in massive plastic pollution becoming an environmental threat worldwide with potential risks for biota and also for human health [1,2]. Therefore, the assessment of the amount, distribution and nature of the plastic debris in ecosystems, and especially in the marine environment, is the focus of intense multidisciplinary research [3,4]. Currently, about 5–13 million tons of plastic waste are estimated to enter the ocean every year [5]. A generally accepted picture based on an increasing number of environmental studies suggests that the largest fraction of it consists of microplastics (MPs) [6]. While there is currently no scientific or regulatory agreement on the definition of MPs size range, the commonly adopted 5 mm as the upper limit [7,8] is being questioned as it relates to the ingestion by fish rather than to the physical and chemical properties. In fact, the latter are mainly related to the specific surface area and its rapid increase (roughly by a half of the third power) with the reduction of particle size; on the other hand, the specific surface area is likely to be more strictly related to the rate and extent of polymer degradation under environmental conditions, and thus with the interaction of MPs with the environment. A scientifically more appropriate size range of 1–1000 μm for MPs has recently been proposed [3,9]. The relevance of the size range of the MPs is exemplified by the observation that the higher the specific surface area, the faster the chemical degradation, mainly due to photo-oxidation processes, leading to the formation of leachable, soluble and/or volatile degradation products that need to be considered as an emerging source of environmental pollution deriving from MPs [10,11,12,13]. The most common analytical techniques used for the analysis of MPs are micro Fourier transform infrared (μ-FTIR), and micro-Raman spectroscopy [14,15], both providing information that does not allow to clearly highlight the differences between the surface and bulk composition and to pinpoint the presence and relative amount of highly degraded fractions and of degradation products that are likely to affect the most the chemical behavior of MPs. On the other hand, in the last few years thermo-analytical approaches have emerged as powerful tools for studying MPs [16,17,18], also in routine analysis of contaminated environmental samples [19]. Among them, analytical pyrolysis [20,21,22] allows the sensitive and accurate characterization of polymers and polymer mixtures through their pyrolytic profiles. The systematic study presented here is based on a combination of analytical pyrolysis approaches and mass spectrometry detection to evaluate the effects of simulated environmental photo-oxidative degradation processes in five different reference MPs. We describe the parallel characterization of the bulk chemical modifications, and of the low-molecular-weight, leachable or soluble organic fraction resulting from the polymer degradation. Investigating the correlation between the modifications of the different polymer types under simulated environmental aging, and the nature and amount of the low molecular weight species resulting from the polymer degradation, is fundamental in order to assess the possible role of such low molecular weight species as emerging environmental pollutants as they may leach out of the MPs, being thus potentially harmful to living organisms. From the most recent literature, a need emerges for studies about the potential harmfulness of the oxidation products (low molecular weight and oxidized oligomeric species) that may leach out from plastic debris dispersed in the environment. The formation of such species is generally neglected in the studies on MPs even if these compounds may pose even higher risks for the environment and the biota than the MP particles themselves, risks that are far from being understood and assessed.

In this context, the aim of this study was the thorough characterization of aged (photo-oxidized) MPs and of their low molecular weight and/or highly oxidized fraction as evaluated by their enhanced solubility in polar organic solvents. Five reference polymers were selected among those most commonly found in the form of MPs polluting the environment, namely: polypropylene (PP), polystyrene (PS), polyethylene terephthalate (PET), low-density polyethylene (LDPE) and high-density polyethylene (HDPE). The virgin polymers in the form of micropowders (average particle size in the 857–509 μm range) were aged in a solar-box for four weeks. Samples were periodically collected and analyzed by means of evolved gas analysis–mass spectrometry (EGA–MS) and the results were compared with those obtained for the unaged polymers in order to study the thermal degradation behavior specific for each polymer during photo-aging. The unaged and the polymer samples aged 4 weeks were also characterized by means of pyrolysis–gas chromatography–mass spectrometry (Py–GC–MS) to detect and identify the produced alteration and degradation products. Finally, all samples were subjected to solvent extraction in either methanol (for PS) or refluxing dichloromethane (for PP, PET, LDPE and HDPE), that are non-solvents for the virgin polymers, to selectively extract the low molecular weight fractions generated by chain scissions as a result of extensive degradation. The solvent extracts were then analyzed by size exclusion chromatography (SEC) and Py–GC–MS; for the latter analyses hexamethyldisilazane (HMDS) derivatization was performed to allow detection of high polarity and low-volatility species such as those resulting from photo-oxidative processes entailing oxygen pickup through free radical reactions. The insoluble polymer fractions were also analyzed by means of Py–GC–MS in an attempt to evaluate the chemical modifications preliminary to the most extensive degradation processes, and compare them with those of the most highly degraded, solvent extractable fractions.

## 2. Materials and Methods

### 2.1. Chemicals

Dichloromethane (DCM, high-performance liquid chromatography (HPLC) grade, Sigma-Aldrich, St. Louis, MO, USA) and methanol (MeOH, HPLC grade, Sigma-Aldrich) were used as solvents in the extractions. Hexamethyldisilazane (HMDS ≥ 99%, Sigma-Aldrich) was used as derivatizing agent for the in situ thermally assisted silylation of the pyrolysis products bearing carboxylic and hydroxyl groups in the Py–GC–MS analysis of the polymer extracts.

### 2.2. Reference Polymers

Micronized polypropylene (PP), polystyrene (PS), polyethylene terephthalate (PET), low-density polyethylene (LDPE) and high-density polyethylene (HDPE), with average particle size in the 857–509 μm range depending on the polymer as reported elsewhere [23,24], were kind gifts from Poliplast S.p.A (Casnigo, Italy).

### 2.3. Artificial Aging and Extraction

PP, PS, PET, LDPE and HDPE micropowders were artificially aged for four weeks using a solar-box system (CO.FO.ME.GRA. Srl, Milan, Italy) equipped with a Xenon-arc lamp and outdoor filter. The conditions for the aging were: temperature 40 °C, irradiance 750 W/m^2^, relative humidity around 60%. Aliquots (ca. 200 mg) of each polymer were collected before (0w) and after 1 (1w), 2 (2w), 3 (3w), and 4 (4w) weeks of artificial aging and stored in sealed glass vials at −20 °C until analysis [23]. About 150 mg of each unaged and aged polymer sample was extracted for 6 h with 30 mL either MeOH (for PS) or DCM (all other polymers) in a Soxhlet apparatus, collecting both the residues and the extractable fractions for the subsequent characterizations. MeOH and DCM, that act as non-solvents for the virgin polymers, were chosen to selectively extract the degraded, low molecular weight fractions. The extracted fractions were dried in a rotary evaporator until constant weight and then stored in glass vials before the Py-GC-MS and SEC analyses. Procedural blanks (DCM and MeOH) were also prepared and analyzed along with the polymer extracts. The overall procedure is schematically summarized in Figure 1.

### 2.4. Analytical Methods and Instrumentation

#### 2.4.1. Evolved Gas Analysis–Mass Spectrometry (EGA–MS)

The EGA-MS analyses of unaged and artificially aged bulk polymers were performed with an EGA/PY-3030D micro-furnace pyrolyzer (Frontier Laboratories Ltd., Koriyama, Japan) coupled to a 6890 gas chromatograph and a 5973 mass spectrometric detector (Agilent Technologies, Santa Clara, CA, USA). The experimental conditions were the following: temperature ramp for the furnace from 50 °C to 700 °C at 10 °C/min; interface between the pyrolysis furnace and the GC–MS system set at a temperature 100 °C higher than that of the furnace but limited to a maximum of 300 °C; GC injector operated in split mode (20:1 ratio) at 280 °C [25,26]. The evolved pyrolysis products were directly sent to the mass spectrometer using a UADTM-2.5N deactivated stainless-steel capillary tube (3 m × 0.15 mm, Frontier Laboratories Ltd., Japan) held at 300 °C, and using helium (1 mL/min) as the carrier gas. The temperature of the transfer line to the mass spectrometer was 280 °C. The mass spectrometer was operated in electronic impact (EI) positive mode (70 eV, *m*/*z* range 15–700). The temperatures of the ion source and quadrupole analyzer were 230 °C and 150 °C, respectively. Each 100–250 μg sample was directly weighed in the deactivated stainless-steel pyrolysis cup with an XS3DU microanalytical scale (Mettler-Toledo, Columbus, OH, USA) with seven digits and a precision of 1 μg.

#### 2.4.2. Pyrolysis–Gas Chromatography–Mass Spectrometry (Py–GC–MS)

Analyses of unaged and artificially aged bulk polymers, their extracts, and the corresponding extraction residues (insoluble fractions) were performed using a multi-shot pyrolyzer EGA/PY-3030D (Frontier Lab.) coupled with a 6890N gas chromatography system with a split/splitless injection port and combined with a 5973 mass selective single quadrupole mass spectrometer (Agilent Technologies). The samples (50–100 μg) were placed in deactivated stainless-steel sample cups. Pyrolysis conditions were optimized as follows: pyrolysis temperatures were selected based on the samples analyzed [25,26,27]; interface 280 °C; GC injector temperature 280 °C; GC injection operated in split mode with an optimized 10:1 split ratio. The chromatographic separation of the pyrolysis products was performed on a fused silica capillary column HP-5MS (5% diphenyl–95% dimethyl-polysiloxane, 30 m × 0.25 mm internal diameter., 0.25 μm film thickness, J&W Scientific, Agilent Technologies) preceded by a deactivated fused silica pre-column (2 m × 0.32 mm i.d.). The chromatographic conditions were: 40 °C for 5 min, 10 °C/min to 310 °C for 20 min, carrier gas (He, 99.9995%) flow 1.2 mL/min. MS parameters: electron impact ionization (EI, 70 eV) in positive mode; ion source temperature 230 °C; scan range 35–700 *m*/*z*; interface temperature 280 °C. Perfluorotributylamine (PFTBA) was used for mass spectrometer tuning. MSD ChemStation (Agilent Technologies) software was used for data analysis and peak assignment was based on mass spectra libraries (NIST 8, score higher than 80%) and literature data [20]. For the analysis of the polymer samples and the polymer residues (after extraction), the pyrolysis furnace was set at 600 °C and samples ranging from 50 to 100 μg were directly weighed in the deactivated stainless-steel pyrolysis cup with an XS3DU microanalytical scale (Mettler-Toledo, USA) with seven digits and a precision of 1 μg. For the analysis of the extracted fraction of the reference polymer samples, 1 mL of DCM was added to each vial containing the polymer dried extracts; regarding PS dried extract, 1 mL of methanol was used, since it is the solvent used for its extraction. Then, different volumes of the extracts (20–340 μL) were directly dried in the pyrolysis cup and then weighed with an XS3DU microanalytical scale in order to obtain about 100 μg of sample. 4 μL of HMDS were added in the pyrolysis cup as derivatizing agent in order to detect polar and low-volatility compounds. Pyrolysis temperature was set at 550 °C. HMDS was used in the cleaning pre-treatment of the Py–GC–MS system.

#### 2.4.3. Size-Exclusion Chromatography (SEC)

Size-exclusion chromatography (SEC) analysis of the extracts of unaged and aged polymers was performed with a Jasco (Jasco Europe srl, Cremella, LC, Italy) instrument comprising a PU-2080 Plus four-channel pump with degasser, two in series PL gel MIXED-E Mesopore (Polymer Laboratories, Church Stretton, UK) columns placed in a Jasco CO-2063 column oven thermostated at 30 °C, a Jasco RI 2031 Plus refractive index detector, and a Jasco UV-2077 Plus multi-channel ultraviolet (UV) 120 spectrometer; the ChromNav Jasco software was used for data acquisition and analysis. The eluent was trichloromethane (HPLC grade Sigma-Aldrich) at 1 mL/min flow rate.

## 3. Results and Discussion

The micropowders of all reference polymers were artificially aged 4 weeks under conditions roughly corresponding to a 6-month exposure at the latitude of the Tuscany region, central Italy. The pristine (unaged virgin polymers) and the irradiated samples collected after each subsequent 1-week period of artificial aging were analyzed by means of EGA–MS and Py–GC–MS to gain information about the extent of the photo-oxidative degradation and the type of chemical damage induced by the photo-aging. Pristine and artificially aged polymers were also extracted with solvents suitable for separating the low molecular weight photo-oxidized fragments from the insoluble bulk polymer. Finally, the two fractions obtained upon solvent extraction of each sample (organic extract containing the soluble degradation products, and residue containing the insoluble polymers) were characterized by Py–GC–MS and, in the case of the extracts containing the soluble fraction produced upon aging, also by SEC. The extraction yields and their variation during artificial aging are reported in Figure 2. The extraction yields increased steadily throughout the investigated aging time for LDPE, PP, and PS. The higher sensitivity of these polymers towards aging can be related to the presence of tertiary and benzyl carbon atoms, which are more prone to be attacked by free radical species directly or indirectly generated by photo-irradiation, and therefore more susceptible to undergo C–C bond cleavage as a result of secondary processes (e.g., β-cleavage of oxy-radicals generated upon decomposition of peroxy-radicals, the latter resulting from oxygen pickup by the primary radicals produced by H-abstraction). However, while the extractable fraction increases linearly with the irradiation time for the three polyolefins (i.e., including HDPE), for PS a tendency resembling an exponential growth or indicating some initial inhibition effect is observed. This result can be interpreted as an effect of possible radical scavenging associated with the presence of the monosubstituted phenyl ring and/or with the higher glass transition temperature (Tg) of PS: a lower diffusional mobility of free radical species and thus an induction time associated with slower increase of their concentration is typically associated with free radical transfer and oxygen pickup.

### 3.1. EGA–MS and Py–GC–MS Analysis of Polymers during Artificial Aging

EGA analyses were performed on the 0w (unaged), 1w, 3w, and 4w polymer samples to evaluate changes in the thermal degradation temperature profiles. The 0w and 4w samples, representing the initial and final situation, were also analyzed by Py–GC–MS. The results are reported and discussed in the following paragraphs, separately for each polymer type.

#### 3.1.1. Polypropylene

The EGA profiles for virgin and aged PP are shown in Figure 3, each curve being the average of five replicates.

Based on a statistical evaluation of the five replicated analyses (*t*-test, 95%), a difference higher than 5.2 °C in the thermal degradation temperature, T_D_, taken as the temperature corresponding to the maximum of the EGA peak, can be considered as significant. The EGA profiles show a change in the thermal degradation profile of PP upon aging. In particular, photodegradation induces a progressive decrease of T_D_ and a broadening of the thermal degradation peak (Table 1), with a T_D_ drop of 35 °C from T_D_ = 453 °C for the unaged PP-0w down to 418 °C for PP-4w. An even more consistent drop is observed for the onset temperature of the thermal degradation peak: from 421 °C for the unaged PP-0w down to 350 °C for PP-4w.

Each average mass spectrum is obtained by averaging the EGA–MS mass spectra in the temperature range corresponding to the peak. The most abundant ion fragments in the average EGA mass spectrum of PP-0w (see Appendix A in the Appendix A; spectra collected in the temperature range from 421 °C to 480 °C) are *m*/*z* 43, 69, 83, 97, 111, 125, 153, corresponding to PP oligomers with different chain lengths (Appendix A in Appendix A) [20].

The progressive shift of TD max towards lower temperatures at increasing aging time is in agreement with the known strong susceptibility of PP to chain scission reactions as a result of photo-induced degradation processes, being PP reactivity higher than that of both HDPE and LDPE, due to the higher concentration of tertiary C–H bonds [10,28,29]. This behavior is in full agreement with the evolution of the EGA profiles during artificial aging as observed in our experiments, featuring the progressive T_D_ lowering and peak broadening discussed above, as a result of the growing concentration of chain scission products and of reactive oxidized and unsaturated groups, less thermally stable than saturated hydrocarbon structures. Such structural changes did not induce any significant change in the EGA–MS mass spectra of the polymer thermal degradation products.

The Py–GC–MS chromatogram of PP-0w is shown in Figure 4, while its peak assignments are listed in Appendix A in Appendix A. The most abundant pyrolysis products are 2,4-dimethyl-1-heptene (n° 6), 2,4,6-trimethyl-1-nonene (n° 12, 13), 2,4,6,8-tetramethyl-1-undecene (n° 16, 17, 18) along with other polypropylene oligomers of increasing chain length (n° 19–44), in agreement with the well-known pyrolytic behavior of PP involving random C–C scissions followed by intramolecular H transfer yielding alkanes and 1-alkenes [20,30]. The pyrolysis profiles of PP-0w and PP-4w are very similar. In particular, no pyrolysis products indicative of the occurrence of photoaging-related polymer degradation could be detected, even if the EGA profile did highlight the occurrence and progress of polymer degradation processes during the aging experiment. The lack of detectable molecular fragments that could be associated with the increasing fraction of oxidized products resulting from the photo-oxidative degradation may be explained by their low concentration in the bulk polymer, and by the poor effectiveness of the GC–MS separation and detection section in a conventional Py–GC–MS apparatus when oxidation products are involved, unless in situ thermally assisted derivatization of the carboxylic/hydroxyl functions is applied prior to the analysis

#### 3.1.2. Polystyrene (PS)

All the EGA–MS curves feature a single peak in the 350–450 °C range (Appendix A in Appendix A), with T_D_ max at about 406 °C. A slight decrease of the onset temperature (ΔT = 8 °C) observed in the EGA profile of the PS-3w and PS-4w samples can be associated to the formation of photo-oxidized products, although in a comparatively smaller amount with respect to the case of PP as seen before. In the average mass spectrum of PS (350–450 °C, Appendix A in Appendix A), the most abundant ions are fragments with *m*/*z* 51, 65, 78, 91, 104, 117, 207, corresponding to the ions in the mass spectra of the well-known pyrolysis products of the polymer (toluene, styrene, styrene dimer and styrene trimer, Appendix A in Appendix A) [20]. By comparing the mass spectra of the four samples, no significant differences in the relative abundance of the main ions are detected. In the Py–GC–MS chromatogram of PS-0w (Figure 5 and Appendix A in Appendix A) the most abundant pyrolysis products are styrene (n° 2), α-methylstyrene (n° 5), 3-butene-1,3-diyldibenzene (styrene dimer, n° 13) and 5-hexene-1,3,5-triyltribenzene (styrene trimer, n° 17). The observed decrease in the onset temperature is in agreement with the typical thermal degradation processes of PS, mainly characterized by depolymerization pathways.

The pyrolysis profile of PS-4w (Appendix A in Appendix A) and PS-0w are very similar. This lack of differences between the two pyrolysis profiles can be explained by the tendency of PS to produce hydroquinonic structures as a result of photo-oxidation, which are known for their antioxidant and free radical scavenging properties and are thus likely to generate a chemically altered surface layer protecting the bulk polymer from further photo-oxidative degradation [28,31]. 

The yellowing observed in the PS-4w sample is a direct consequence of such surface-limited formation of oxidized and possibly conjugated aromatic structures, which do not significantly affect the EGA–MS curves and the pyrolysis profiles of the aged PS because degradation only involves a small fraction of the overall polymer mass.

#### 3.1.3. Polyethylene Terephthalate (PET)

The EGA profile of the unaged PET-0w sample (Appendix A in Appendix A) shows a peak from 381 °C to 463 °C with a maximum (T_D_) at 410 °C. The EGA profiles of the artificially aged samples do not show any significant variation in the T_D_ or in the relative abundance of the main ions in the average mass spectrum (381–463 °C, Appendix A in Appendix A). The latter are fragments with *m*/*z* 44, 77, 105, 122, 149, 297 which correspond to the ions in the mass spectra of the thermal degradation of the polymer: vinyl benzoate, benzoic acid, divinyl terephthalate, and 2-(benzoyloxy) ethyl vinyl terephthalate. As in the case of PS, a slight decrease of the onset temperature (ΔT = 7 °C) is only observed in the EGA profile of the PET-4w sample, that can be related to the formation of photo-oxidation products; an extended artificial aging time would be necessary to better investigate the degradation behavior.

The chromatogram obtained in the Py–GC–MS analysis of the unaged PET-0w is shown in Figure 6, while peak identification is listed in Appendix A in Appendix A. The main pyrolysis products of PET are vinyl benzoate (n° 9), benzoic acid (n° 10), biphenyl (n° 11), divinyl terephthalate (n° 12) and ethan-1,2-divinyldibenzoate (n° 18). The Py–GC–MS profile of the PET-4w sample (Appendix A in Appendix A) is also in this case very similar to that of the unaged polymer, in agreement with its well-known higher photostability compared to polyolefins [24,28]. In the Py–GC–MS profile of the artificially aged polymer only a slight increase of the relative abundance of acetophenone, benzaldehyde, vinyl benzoate, dibenzofuran, and fluorenone is observed.

#### 3.1.4. Polyethylene (PE)

Both HDPE and LDPE were investigated, and are both discussed in this section. The EGA profile of the unaged (Appendix A in Appendix A) LDPE-0w sample shows a peak with T_D_ at 454 °C; artificial aging induces a slight increase of the baseline and a shift (ΔT = 7 °C) of the onset to lower temperatures, indicative of the formation of oxidation products. No significant changes are observed in the average mass spectra of the artificially aged LDPE samples (Appendix A in Appendix A). The main ions are the fragments with *m*/*z* 43, 57, 69, 83, 97, 111, 125, 139, 154 which correspond to polyethylene oligomers of different chain lengths (Appendix A) [20]. The EGA curves of the unaged and aged HDPE samples (Appendix A in Appendix A) show a peak with T_D_ at 474 °C, 20 degrees higher than that recorder for LDPE samples. Even in this case, artificial aging induces a slight decrease of the onset temperature (ΔT = 6 °C) probably due to the presence of oxidation products at low concentration as a result of photo-oxidative degradation. The average mass spectra from the EGA profiles of the HDPE samples (Appendix A in Appendix A) are equivalent to the LDPE ones. The results obtained in the Py–GC–MS of the LDPE-0w are reported in Figure 7 and Appendix A in Appendix A. The pyrogram features a series of clusters comprising three main peaks each, assigned to the diene (most likely an α,ω-diene, C_n:2_), the monoalkene (most likely a 1-alkene, C_n:1_), and the alkane, respectively, for any given C_n_ hydrocarbon in the chain lengths range C_6_–C_26_. At the right of each cluster, the peak corresponding to the C_n-2_ linear aldehyde is observed, with low relative intensity.

After artificial aging, new peaks next to the triplets are observed in the pyrolytic profiles of both HDPE and LDPE; these can be associated to oxidation products and in particular to linear ketones, linear saturated alcohols, and monocarboxylic acids. Peaks corresponding to saturated aldehydes, with lengths up to C24, increase in their relative intensity with aging (Figure 8) [32]. The complete list of all the pyrolysis products detected in the chromatogram obtained in the Py–GC–MS analysis of LDPE-4w is reported in Appendix A in the Appendix A. The oxidation products detected in the chromatogram can either be the products of the thermolytic cleavage of mildly oxidized high molecular weight polymer chains, or smaller oxidized fragments produced by photolytic oxidation and chain scission as a result of photo-oxidative artificial aging, or both. Even though the general features of the pyrolysis profiles of the two aged polymers are very similar, in the case of LDPE a slightly higher number of oxidized pyrolysis products were detected compared to HDPE, probably due to the different intermolecular forces in the structures of the two polymers and consequence differences in the ageing behaviors. Appendix A report the complete list of the pyrolysis products identified in the chromatogram obtained in the Py-GC-MS analysis of the HDPE-0w and HDPE-4w samples, respectively.

### 3.2. Analysis of Extractable Fraction of Reference Polymers before and after Artificial Aging

The results obtained suggest that low molecular weight degradation and oxidation products are not detectable by Py–GC–MS analysis of the aged polymer, because the peaks deriving from the unaltered fraction of the polymer hinder the detection of low-intensity peaks associated to the lower fraction of altered portion of the polymer. The alteration induced on the five reference plastics by the photoaging was thus investigated also by a complementary approach: the aged samples were extracted with polar organic solvent and the composition of the extracts was investigated by SEC and Py–GC–MS, and compared with the extracts obtained from the unaged corresponding polymer. PP, PET, HDPE, and LDPE were subjected to DCM extraction, while MeOH was used for PS. Procedural blanks were also prepared for comparison. Py–GC–MS analysis of the extractable fractions allowed us to achieve an enhanced sensitivity towards the degradation products, focusing on the extractable and leachable components to gain additional information on the degradation processes occurring during the aging of MPs. Py–GC–MS was carried out with the addition of HMDS in order to detect and characterize the low-volatile and polar compounds such as aldehydes, alcohols and carboxylic acids strictly related to photo-oxidative degradation; HMDS also achieves the derivatization of PET pyrolysis products.

#### 3.2.1. Polypropylene

The chromatogram obtained in the Py(HMDS)–GC–MS analysis of the DCM extract of the PP-4w sample is reported in Figure 9, with peak identification in Table 2. The pyrolytic profile shows oxidized products such as different chain length mono- and dicarboxylic acid in the first part of the chromatogram (14–24 min). In particular, low-molecular weight differently branched monocarboxylic acids are detected in the C_2_–C_6_ range, such as butanoic acid (n° 5), 2-butenoic acid (n° 6), 2-methyl-4-pentenoic acid (n° 8), 3-methyl-3-butenoic acid (n° 9), 2-hydroxypropanoic acid (n° 12), hydroxyacetic acid (n° 13) 2-hydroxy-2-propenoic acid (n° 15), 2-hydroxybutanoic acid (n° 16), 4-oxopentanoic acid (n° 17), 3-hydroxypropanoic acid (n° 18), 3-hydroxybutanoic acid (n° 19), and 3-hydroxy-3-butenoic acid (n° 20). Low molecular weight dicarboxylic acids such as butanedioic acid (n° 22) and methylbutanedioic acid (n° 23) are also observed in the first part of the profile. The second part of the chromatogram (24–35 min) is mainly characterized by different chain-length PP oligomers that are soluble in DCM.

The results show the presence of oxidation products that could not be detected in the pyrogram of the aged bulk sample analyzed without in situ thermally assisted derivatization with HMDS of the carboxylic/hydroxyl functions, due both to their low concentration in the bulk polymer, and to their polarity—low volatility—that prevented their GC separation as such. The carboxylic and dicarboxylic acids observed in the Py–GC–MS analysis of the extract of artificially aged PP-4w are not detected in the extract of unaged PP-0w (Appendix A in Appendix A), clearly indicating that they are the result of photo-oxidative degradation rather than of pyrolytic fragmentation. Figure 10 reports the SEC chromatograms (10–25 min) obtained for the DCM extracts of PP-0w (red), PP-1w (green), PP-3w (yellow), and PP-4w (blue). The profiles show two peaks at high retention times (about 20.4 min and 22.5 min) corresponding to low molecular weight fractions. This is expected since high molecular weight polyolefins are insoluble in DCM. No significant differences are highlighted when comparing the SEC profiles of the unaged and aged PP extracts.

#### 3.2.2. Polystyrene

The chromatogram obtained in the Py(HMDS)–GC–MS analysis of the MeOH extract of PS-4w is reported in Figure 11. Peak identification is in Table 3. The main pyrolysis products are the same as those observed in the Py–GC–MS chromatogram of PS-4w analyzed in “bulk” (non-subjected to extraction, Figure 5): styrene (n° 2), benzoic acid (n° 26), its dimer (n° 39), and its trimer (n° 53). Different acids and alcohols deriving from benzoic acid are detected, like 4-methylphenol (n° 20), 1-phenylethenol (n° 25), 3-methylphenol (n° 27), phenylacetic acid (n° 28), phenylpropanoic acid (n° 32), and 4-hydroxybenzoic acid (n° 36); dicarboxylic acids and other carboxylic acids are also found: 2-hydroxy-propanoic acid (n° 14), hydroxyacetic acid (n° 16), 4-hydroxy pentanoic acid (n° 17), 3-hydroxypropanoic acid (n° 19), butanedioic acid (n° 29), methylbutanedioic acid (n° 30), 1,4-benzenedicarboxylic acid (n° 42), pentadecanoic acid (n° 48), hexadecenoic acid (n° 51), and octadecanoic acid (n° 52).

None of the oxidized compounds identified in the pyrolysis profile of the MeOH extract of PS-4w are detectable in the extract of the PS-0w (Appendix A in Appendix A), indicating that the former are the result of extensive photo-oxidative degradation occurred during artificial aging. Figure 12a,b report the SEC chromatograms (10–27 min.) of the MeOH extracts of PS-0w (red), PS-1w (green), PS-2w (purple), PS-3w (yellow), PS-4w (blue), acquired at 260 nm and 340 nm, respectively.

The extracts of the aged PS samples show a broad structured band in the 12–20 min range, which corresponds to fractions with molecular weight ranging from 10,200 Da to values below 500 Da; whereas the profile of the extract of the PS-0w (red) appears less-intense in the 260 nm chromatogram, and it is almost non-detectable in the 340 nm chromatogram, showing a band in the 16–19 min range. Aging induces the disappearance of the peak at 16 min (black arrow) and the appearance of a new one at about 19 min (dotted black arrow) which corresponds to low-molecular weight fraction (below 500 Da). 

#### 3.2.3. Polyethylene Terephthalate

The chromatogram obtained in the Py(HMDS)–GC–MS analysis of the DCM extract of the PET-4w sample is reported in Figure 13, with peak identification in Table 4. The pyrograms of the DCM extract of PET-0w (Appendix A in Appendix A) and PET-4w samples are very similar, in agreement with the results obtained by EGA–MS and Py–GC–MS for the bulk (non-extracted) polymer, highlighting PET photo-oxidative stability. In particular, the typical pyrolysis products of the polymer are observed: benzoic acid (n° 14), vinyl benzoate (n° 22), and divinyl terephthalate (n° 24). The identified carboxylic acids and benzenedicarboxylic acids—hydroxybenzoic acid (n° 11), 3-phenyl-2-propenoic acid (n° 20) and 1,4-benezenedicarboxylic acid (n° 26)—were also present in the pyrogram of the DCM extract of the unaged sample, suggesting that they are not the result of photo-oxidative degradation but more likely they are the result of thermolytic cleavage and rearrangements occurring during pyrolysis, and that irradiation of PET microparticles did not produce a significant amount of extractable oxidation products, highlighting the stability of PET in the adopted conditions.

The PET extracts were not analyzed by means of SEC since the amount of the extractable fraction and the pyrolysis profiles of the extracts of PET-0w and PET-4w samples were nearly identical. The profile observed in the analysis of the extract of the artificially aged PET is essentially the same as that observed for the DCM-soluble fraction of the unaged polymer, and thus the contribution from polymer degradation processes can be assumed to be negligible.

#### 3.2.4. Polyethylene

The DCM extracts of LDPE and HDPE, 0w and 4w, were analyzed by Py(HMDS)–GC–MS. The chromatogram obtained from LDPE-4w is shown in Figure 14, while the main pyrolysis products are listed in Appendix A of the Appendix A.

Saturated monocarboxylic acids, monounsaturated monocarboxylic acids, and dicarboxylic acids are detected in the Py(HMDS)–GC–MS analysis of the extract of the aged sample, together with the pyrolysis products that were observed in the analysis of the unaged bulk (non-extracted) polymer: α,ω-dienes, 1-alkenes and alkanes. These oxidized products are not observed in the Py–GC–MS profile of the extract of the LDPE-0w (in Appendix A, see Appendix A along with the list of all its main pyrolysis products in Appendix A). The complete lists of the pyrolysis products detected in the Py(HMDS)–GC–MS chromatogram of the extracts of the HDPE-0w and HDPE-4w are reported in the Appendix A in Appendix A and Appendix A, respectively. The Py(HMDS)–GC–MS profiles of the pyrograms for the extracts of LDPE-4w and HDPE-4w show very similar general features; however, in the case of aged LDPE a higher number of oxidized pyrolysis products is observed. In particular, the analysis of the extract of the LDPE-4w highlights saturated monocarboxylic acids in the C_4_–C_35_ range, monounsaturated monocarboxylic acids in the C_4_–C_23_ range, and dicarboxylic acids in the C_4_–C_18_ range. For the extract of the HDPE-4w sample, saturated monocarboxylic acids in the Py–GC–MS chromatograms are in the C_5_–C_20_ range, monounsaturated monocarboxylic acids in the C_4_–C_18_ range, and dicarboxylic acids in the C_4_–C_12_ range (Appendix A). Even in this case a recurrent pyrolysis pattern can be highlighted (Figure 15): along with the formation of the α,ω-diene and 1-alkene, we observe the presence of the dicarboxylic acid (C_n-9_), the monounsaturated monocarboxylic acid (C_n-4_) and the monocarboxylic acid (C_n-4_) of a given hydrocarbon (C_n_).

Figure 16a,b show the SEC chromatograms (10–24 min.) of the DCM extracts of HDPE-0w (red), HDPE-1w (green), HDPE-2w (purple), HDPE-3w (yellow), and HDPE-4w (blue) acquired with a RI detector and at 260 nm, respectively. The SEC chromatograms acquired with a RI detector of the HDPE samples show two peaks at around 20 and 22 min corresponding to the low molecular weight fractions that are expected, as for PP, due to the fact that the high molecular weight polyolefins are insoluble in DCM. All the extracts show two signals in the 260 nm chromatographic profiles that are not observed in the chromatograms acquired at 340 nm: the first one at about 10 min, corresponding to the one found in the RI chromatograms, and a second one at about 16 min characterized by low intensity that has no corresponding peak in the RI chromatogram, which suggests the presence of impurities. As for LDPE, the SEC-RI chromatograms are completely identical to the ones of the HDPE extracts, while the 260 nm SEC-UV chromatographic profiles do not show any peak.

### 3.3. Analysis of Insoluble Fractions of Reference Polymers before and after Artificial Aging

The insoluble residues from the DCM (MeOH for PS) extractions were analyzed by means of Py–GC–MS to complement the analysis of the aged polymers and of the extracts, and to evaluate the effectiveness of the extraction procedure.

#### 3.3.1. Polypropylene

The Py–GC–MS profiles of the extraction residues (DCM insoluble fraction) of PP-0w (red) and PP-4w (blue) are reported in Figure 17. The list of the pyrolysis products identified in both the chromatograms is in Appendix A in the Appendix A. The two pyrolytic profiles do not show significant differences, and both feature the usual fragmentation pattern produced in the pyrolysis of PP, indicating that the DCM extraction has proven to be very effective. The extractable fraction of PP is observed to increase with aging: for the unaged PP-0w, the extractable fraction is 0.6% of the sample, while for PP-4w, it reaches 16.3%. PP oligomers with chain length up to C40 are detected in the Py–GC–MS analysis of the extracts, whereas in the pyrolytic profiles of the bulk (non-extracted) PP samples only PP oligomers up to C34 were detected. The usefulness of the extraction procedure is demonstrated by the fact that the oxidized fraction, which cannot be detected in the pyrogram of the aged bulk sample, is detected in the pyrogram of the corresponding extract, allowing the identification of higher molecular weight PP oligomers in the pyrogram of the corresponding extract residue.

#### 3.3.2. Polystyrene

The Py–GC–MS profiles of the extraction residues (MeOH insoluble fraction) of PS-0w (red) and PS-4w (blue) are reported in Figure 18. The complete list of the pyrolysis products identified in both chromatograms in Appendix A in the Appendix A. The pyrolytic profiles of the extraction residues of the unaged PS-0w and aged PS-4w samples are very similar, showing mainly the pyrolysis products of reference unaged PS: toluene, styrene and its dimer and trimer. By comparing the two profiles, the only difference is the lower relative abundance of the styrene dimer and styrene trimer in the profile of the aged PS-4w. After MeOH extraction, the insoluble residue of the yellowed PS-4w becomes colorless, indicating that the colored degradation products formed during artificial aging were efficiently extracted in MeOH.

#### 3.3.3. Polyethylene Terephthalate

The Py–GC–MS profiles of the extraction residues (DCM insoluble fraction) of PET-0w and PET-4w resulted in being completely undistinguishable, as observed also for the DCM extracts. They are also very similar to the pyrograms of the corresponding bulk (non-extracted) samples featuring benzene, vinyl benzoate, benzoic acid, biphenyl, divinyl terephthalate and ethan-1,2-diyldibenzoate. As expected, PET appears to be only mildly affected by photo-oxidation under the adopted conditions. Such higher stability is the result of the absence of labile tertiary C–H bonds and of the prevalence of aromatic carbon atoms in the polymer structure. Appendix A in the Appendix A reports the chromatograms obtained in the Py–GC–MS analysis of the extraction residues of the PET-0w (red) and PET-4w (blue). The complete list of the pyrolysis products identified in both chromatograms is reported in Appendix A in the Appendix A.

#### 3.3.4. Polyethylene

The Py–GC–MS profiles of the extraction residues (DCM insoluble fraction) of LDPE, both unaged 0w and aged 4w, are very similar to those of the corresponding profiles for HDPE. Therefore, only LDPE is discussed here. Figure 19 reports the chromatograms obtained in the Py–GC–MS analysis of the extraction residues of the LDPE-0w (red) and LDPE-4w (blue). The complete list of the pyrolysis products identified in both the chromatograms is reported in Appendix A in the Appendix A. The pyrogram of the extraction residue of LDPE-4w does not show the presence of any of the oxidized products (saturated aldehydes, monocarboxylic acids, alcohols, and ketones) that were observed in the analysis of the extract. DCM has proven to be as effective in the extraction of samples HDPE-4w and LDPE-4w as in the case of samples PP-4w and PS-4w. The extractable fraction increases with aging for both LDPE (from 2.1% for LDPE-0w to 11.3% for LDPE-4w) and HDPE (from 0.6% for HDPE-0w to 2.6% for HDPE-4w). Moreover, even in this case, in the pyrograms of the insoluble residues of both samples PE oligomers are observed to present higher molecular weight (up to C_33_) than in the pyrograms of the corresponding bulk samples (molecular weight up to C_26_).

## 4. Conclusions

The results of this paper demonstrate the potential of EGA–MS, Py–GC–MS, and SEC to provide a comprehensive overview of the effects of aging on the most common synthetic polymers. EGA–MS analyses allowed us to assess the changes in the thermal behavior of bulk polymers induced by photo-oxidative degradation. Py–GC–MS and Py(HMDS)–GC–MS analyses provided information on the degradation products formed during induced aging, even when the degradation products contained highly polar functional groups. The pyrolytic profiles of the insoluble extraction residues of unaged polymers were nearly identical to those of the corresponding insoluble fractions of polymers aged 4 weeks. This demonstrates the effectiveness and the necessity of a solvent extraction step to selectively investigate photo-oxidation products. Finally, SEC analyses allowed us to correlate the extent of oxidation and the reduction of molecular weight due to photo-oxidative degradation.

The stability of polyolefins towards aging was found, as expected, to be strongly related to their structure, and in particular to the number of tertiary carbon atoms, due to the higher stability of the radicals resulting from hydrogen abstraction. Thus, LDPE was more susceptible to degradation than HDPE due to the higher branching, and PP was significantly more susceptible than both types of PE. The main low molecular weight photo-degradation products of polyolefins were long chain alcohols, aldehydes, ketones, carboxylic acids, and hydroxy acids, which were observed in the soluble fractions of aged samples. EGA–MS profiles of PP showed the most marked decrease in the average molecular weight of the polymer chains as the aging time increased.

The two investigated aromatic polymers showed different behaviors. PS was significantly degraded by artificial aging. The main low molecular weight photo-degradation products were alcohols and carboxylic acid, the most abundant being benzoic acid and 1,4-benzenedicarboxylic acid. Interestingly, cross-linking was also observed as a consequence of aging, as highlighted by the SEC analyses; again, this may be the result of the higher stability of the generated free radicals that are more likely to undergo biradical coupling as their concentration in the polymeric material increases. Finally, PET showed the highest stability towards aging, as very small differences were observed comparing fresh and aged samples.

The described experimentation paves the ground for a more thorough and detailed characterization of the oxidation products leaching out from plastic debris dispersed in the environment, which may provide a better understanding of the mechanisms of interaction of MPs with the biosphere as they are likely to go well beyond the simple interaction of biota with solid polymer particles. Future studies should aim at investigating the possible connections between the pollution by microplastics and the toxicity potential of the small, oxidized species that may leach out of MPs in into the environment.

## Figures and Tables

**Figure 1 polymers-13-01997-f001:**
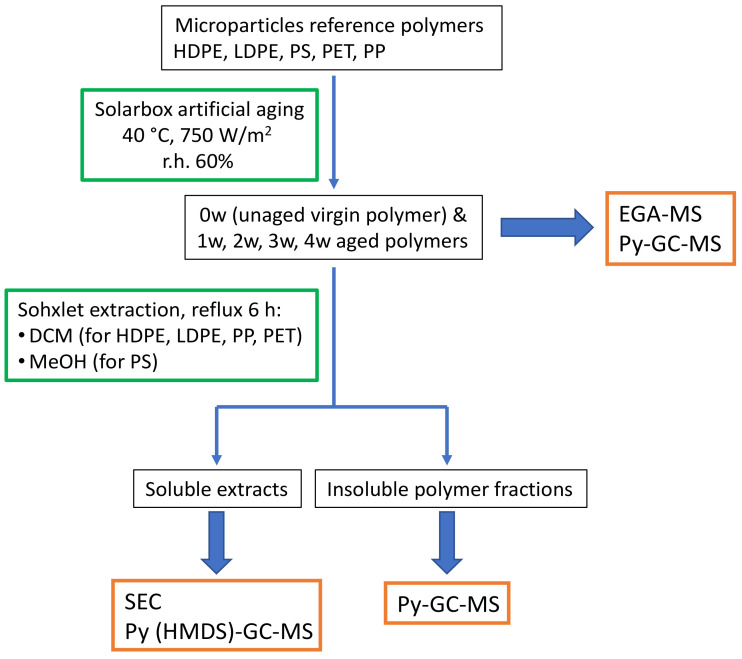
Flowsheet of the overall procedure for the investigation of the polymer degradation products generated upon accelerated photo-oxidative aging.

**Figure 2 polymers-13-01997-f002:**
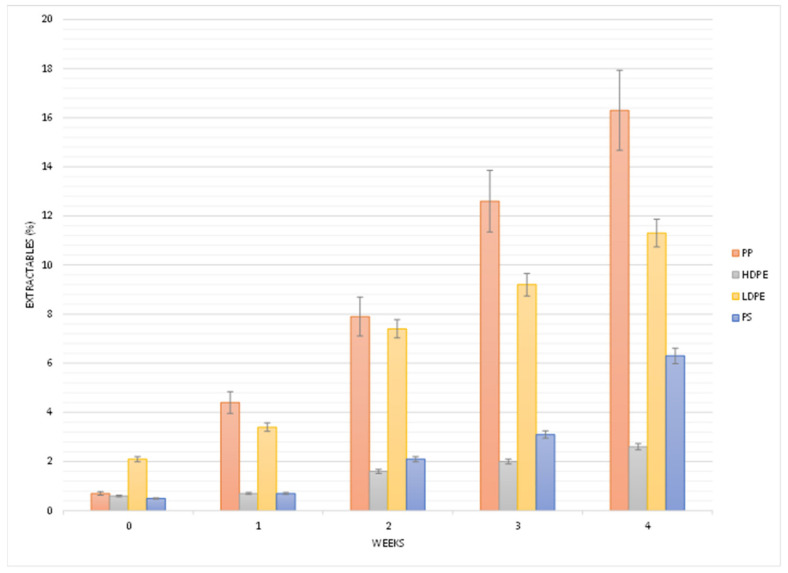
Extractable fractions (w%) for each reference polymer sample upon artificial aging (10% error bars).

**Figure 3 polymers-13-01997-f003:**
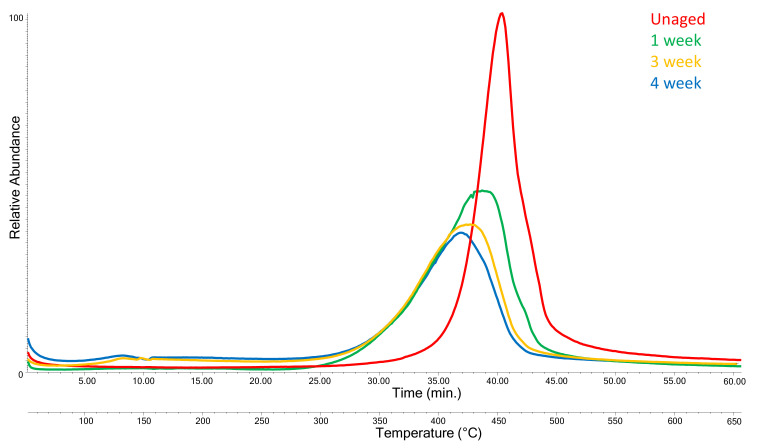
Evolved gas analysis (EGA) profiles of polypropylene (PP) after different artificial aging times: PP-0w (red), PP-1w (green), PP-3w (yellow), and PP-4w (blue) accelerated aging.

**Figure 4 polymers-13-01997-f004:**
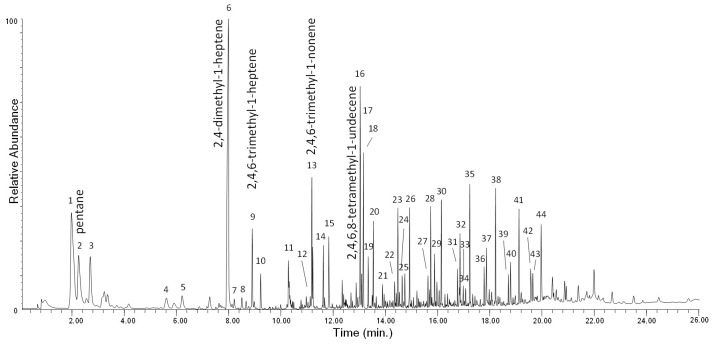
Chromatogram obtained in the pyrolysis–gas chromatography–mass spectrometry (Py–GC–MS) analysis of the unaged PP (PP-0w). Peak identification is reported in Appendix A in Appendix A.

**Figure 5 polymers-13-01997-f005:**
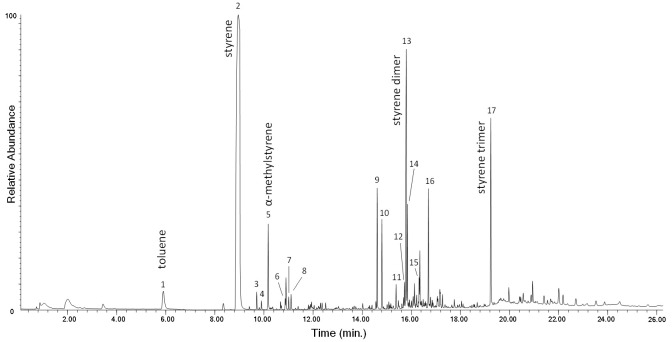
Chromatogram obtained in the Py–GC–MS analysis of the unaged polystyrene (PS-0w). Peak identification is reported in Appendix A in Appendix A.

**Figure 6 polymers-13-01997-f006:**
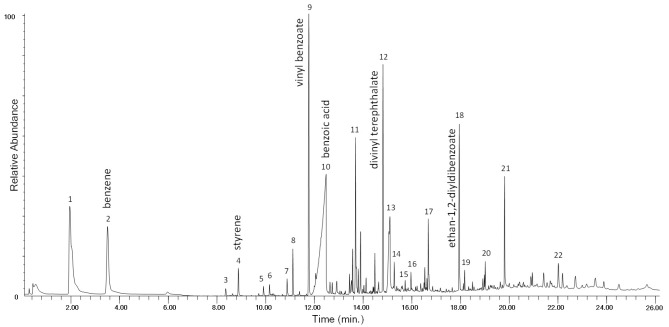
Chromatogram obtained in the Py–GC–MS analysis of the unaged polyethylene terephthalate (PET-0w). Peak identification is reported in Appendix A in Appendix A.

**Figure 7 polymers-13-01997-f007:**
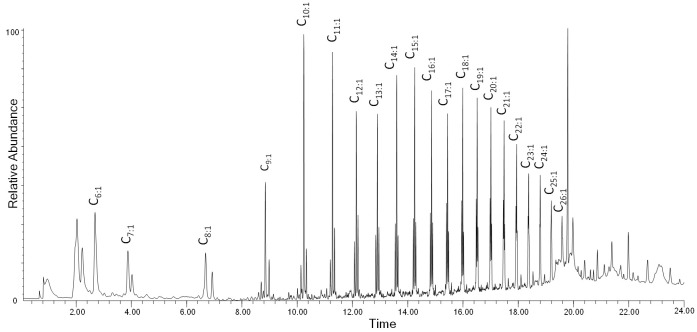
Chromatogram obtained in the Py–GC–MS analysis of the unaged low-density polyethylene (LDPE-0w). C_n:1_ refers to 1-alkenes of a given C_n_ hydrocarbon. Peak identification is reported in Appendix A in Appendix A.

**Figure 8 polymers-13-01997-f008:**
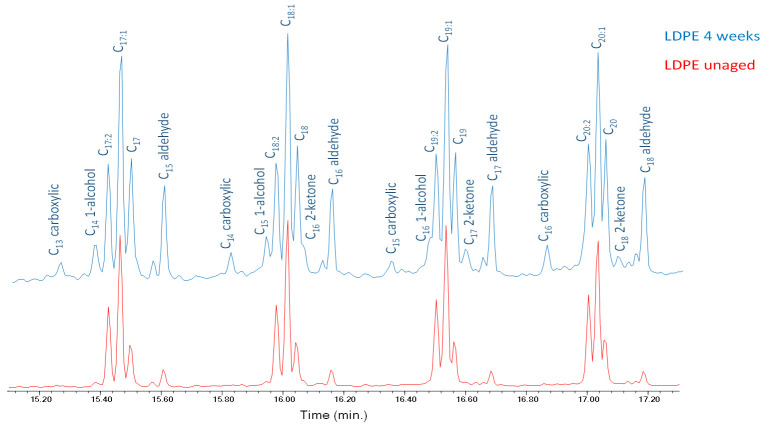
Chromatograms (15.20–17.20 min) obtained in the Py–GC–MS analysis of LDPE-0w (red) and LDPE-4w (blue). C_n:2_ refers to α,ω-dienes, C_n:1_ to 1-alkenes, and C_n_ to alkanes of a hydrocarbon with n carbon atoms.

**Figure 9 polymers-13-01997-f009:**
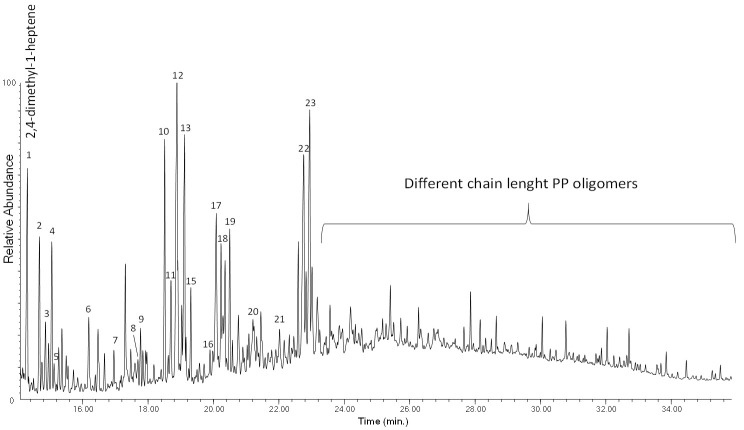
Chromatogram obtained in the Py(HMDS, hexamethyldisilazane)–GC–MS analysis of the dichloromethane (DCM) extract of PP-4w. Peak identification is reported in Table 2.

**Figure 10 polymers-13-01997-f010:**
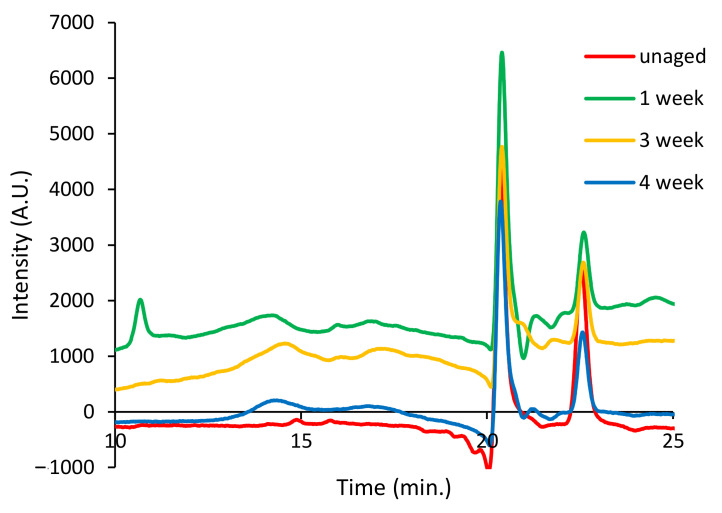
Size exclusion chromatography (SEC) chromatograms (10–25 min) of the DCM extracts of PP-0w (red), PP-1w (green), PP-3w (yellow), and PP-4w (blue); refractive index detector was used.

**Figure 11 polymers-13-01997-f011:**
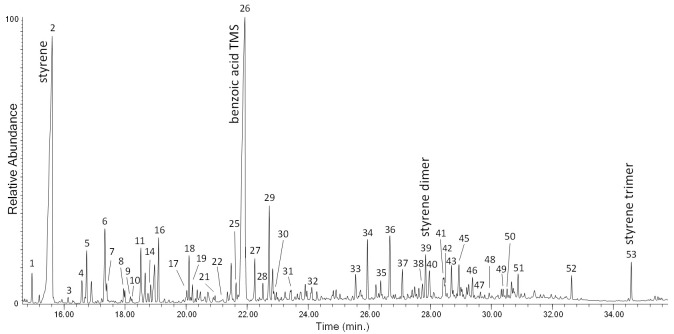
Chromatogram obtained in the Py(HMDS)–GC–MS analysis of the MeOH extract of PS-4w. Peak identification is reported in Table 3.

**Figure 12 polymers-13-01997-f012:**
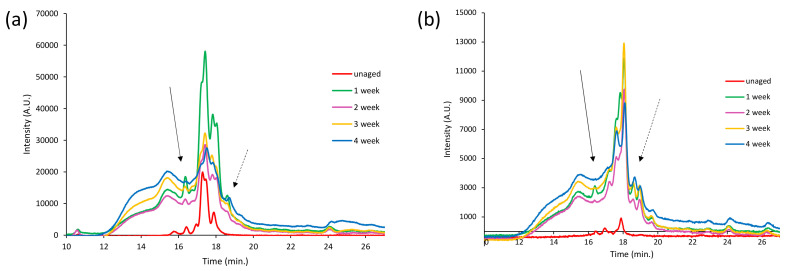
SEC chromatograms (10–27 min) of the MeOH extracts of PS-0w (red), PS-1w (green), PS-2w, PS-3w (yellow), and PS-4w (blue) acquired at 260 nm (**a**) and 340 nm (**b**).

**Figure 13 polymers-13-01997-f013:**
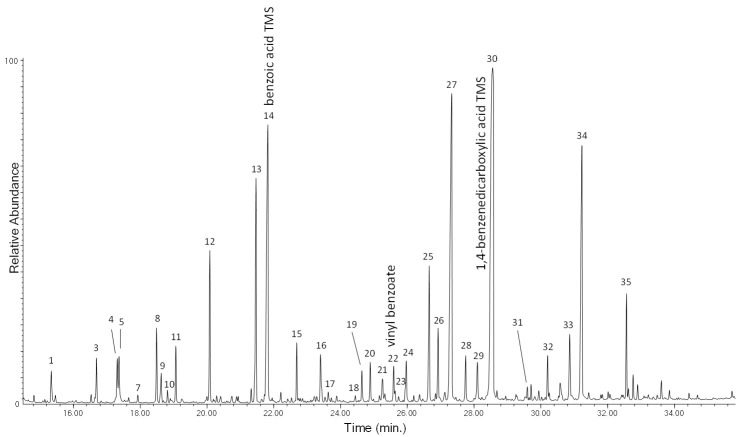
Chromatogram obtained in the Py(HMDS)–GC–MS analysis of the DCM extract of PET-4w. Peak identification is reported in Table 4.

**Figure 14 polymers-13-01997-f014:**
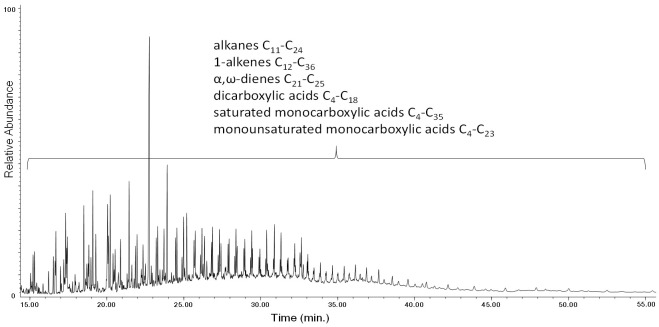
Chromatogram obtained in the Py(HMDS)–GC–MS analysis of the DCM extract of LDPE-4w. The list of the main pyrolysis products is reported in Appendix A in Appendix A.

**Figure 15 polymers-13-01997-f015:**
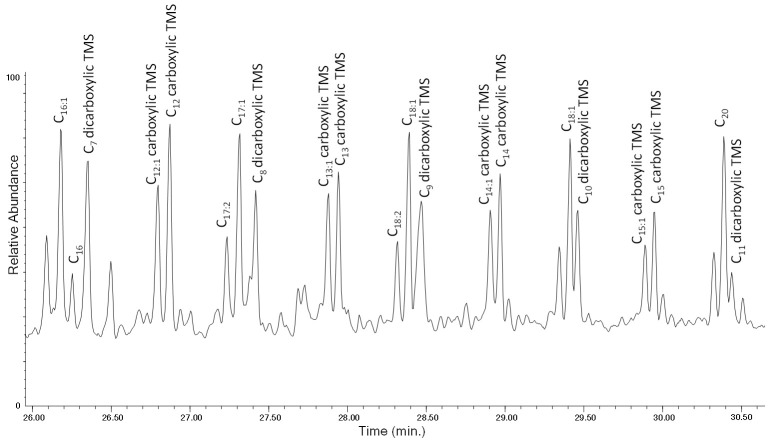
Chromatogram obtained (26.00–30.50 min) in the Py(HMDS)–GC–MS analysis of the DCM extract of LDPE-4w. C_n:2_ refers to α,ω-dienes, C_n:1_ to 1-alkenes, and C_n_ to alkanes of a given C_n_ hydrocarbon.

**Figure 16 polymers-13-01997-f016:**
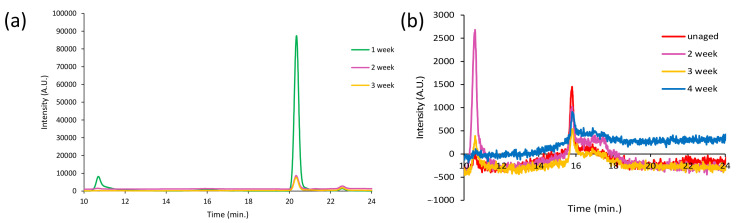
SEC chromatograms (10–24 min) of the DCM extracts of HDPE-0w (red), HDPE-1w (green), HDPE-2w (purple), HDPE-3w (yellow), and HDPE-4w (blue) acquired with a refractive index (RI) detector (**a**) and at 260 nm (**b**).

**Figure 17 polymers-13-01997-f017:**
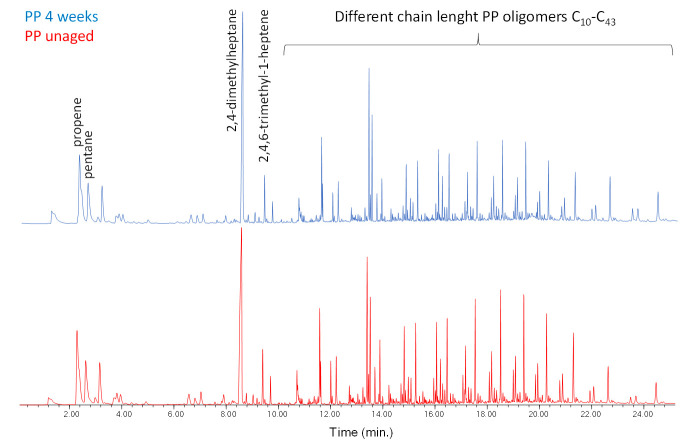
Chromatograms obtained in the Py–GC–MS analysis of the extraction residues of PP-0w (red) and PP-4w (blue). The complete list of the main pyrolysis products is reported in Appendix A in the Appendix A.

**Figure 18 polymers-13-01997-f018:**
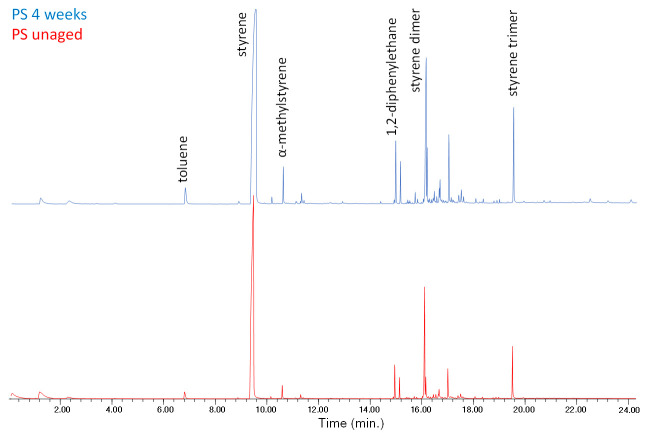
Chromatograms obtained in the Py–GC–MS analysis of the extraction residues of PS-0w (red) and PS-4w (blue). The complete list of the main pyrolysis products is reported in Appendix A in Appendix A.

**Figure 19 polymers-13-01997-f019:**
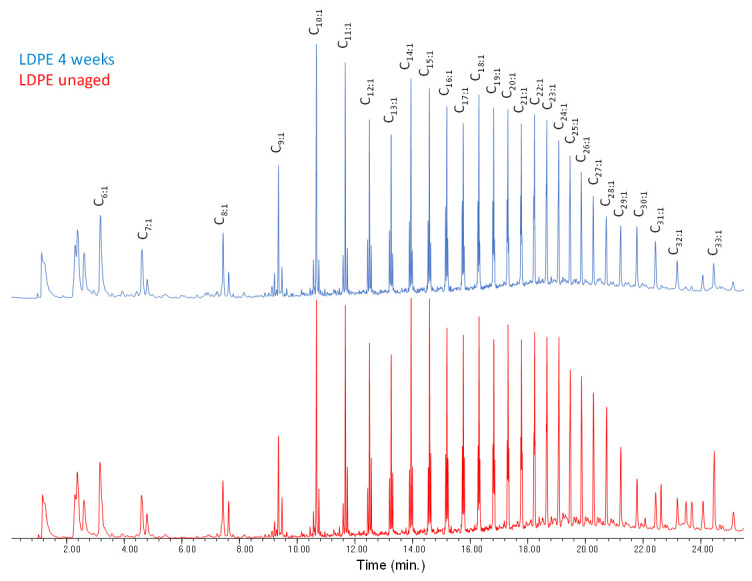
Chromatograms obtained in the Py–GC–MS analysis of the extraction residues of the LDPE-0w (red) LDPE-4w (blue). The complete list of the main pyrolysis products is reported in Appendix A in the Appendix A.

**Table 1 polymers-13-01997-t001:** Degradation temperature ranges and peak maxima (T_D_ max) obtained from the evolved gas analysis–mass spectrometry (EGA–MS) profiles of the PP samples.

Polypropylene	Degradation Temperature Range (°C) ^(a)^	T_D_ Max (°C)
PP-0w	421–480	453
PP-1w	350–484	437
PP-3w	350–463	424
PP-4w	350–458	418

^(a)^ onset and offset temperatures determined at the intersection of the baseline with the tangent at the inflection point of the upward and downward slope of the EGA peak, respectively.

**Table 2 polymers-13-01997-t002:** Identification of peaks in the chromatogram obtained in the Py(HMDS)–GC–MS analysis of the DCM extract of PP-4w (Figure 9). Bold: most abundant species in the chromatogram; most abundant ions in the mass sppectra.

No.	t_r_ (min)	Peak Identification	Main Ions (*m*/*z*)
1	14.3	**2,4-dimethyl-1-heptene**	126, 83, **70**, 55
2	14.7	**xylene**	106, **91**
3	14.9	ethoxytriethylsilane	131, 103, **73**
4	15.05	**octamethyltrisiloxane**	**221**, 73
5	15.1	butanoic acid, trimethylsilyl ester	145, 117, **75**
6	16.2	2-butenoic acid, tert-butyldimethylsilyl ester	**143**, 99, 75, 59
7	16.9	1,2,3-trimethylbenzene	120, **105**
8	17.6	4-pentenoic acid, 2-methyl, trimethylsilyl ester	186, 171, 157, 117, **73**
9	17.8	3-butenoic acid,3-methyl, trimethylsilyl ester	172, 157, 127, 113, **73**, 54
10	18.5	**methyltris(trimethylsiloxy)silane**	295, **207**, 191, 73
11	18.7	unknown	171, 146, 133, 117, **73**
12	18.9	**propanoic acid, 2-[(trimethylsilyl)oxy]-, trimethylsilyl ester**	233, 129, 191, **147**, 133, 117, 73
13	19.1	**acetic acid, [(trimethylsilyl)oxy]-, trimethylsilyl ester**	205, 190, 161, **147**, 133, 117, 103
14	19.2	unknown	171, 157, 145, 129, 117, 103, **75**
15	19.3	2-propenoic acid, 2-[(trimethylsilyl)oxy]-, trimetylsilyl ester	217, **147**, 131, 73
16	19.9	butanoic acid, 2-[(trimethylsilyl)oxy]-, trimethylsilyl ester	233, 205, 190, 147, **131**, 73
17	20.1	**pentanoic acid, 4-oxo-, trimethylsilyl ester**	173, 155, 145, 131, **75**
18	20.2	**propanoic acid, 3-[(trimethylsilyl)oxy]-, trimethylsilyl ester**	219, 177, **147**, 133, 116, 73
19	20.5	**butanoic acid, 3-[(trimethylsilyl)oxy]-, trimethylsilyl ester**	223, 191, **147**, 130, 117, 73
20	21.2	3-butenoic acid,3-(trimethylsilyloxy)-,trimethylsilyl ester	231, 157, 147, **73**
21	22.1	malic acid, O-(trimethylsilyl)-, bis(trimethylsilyl) ester	245, 233, 147, **73**
22	22.7	**butanedioic acid, bis(trimethylsilyl) ester**	247, **147**, 129, 73
23	22.9	**butanedioic acid, methyl-, bis(trimethylsilyl) ester**	261, 217, **147**, 129, 73

**Table 3 polymers-13-01997-t003:** Identification of peaks in the chromatogram obtained in the Py(HMDS)–GC–MS analysis of the MeOH extract of PS-4w (Figure 11). Bold: most abundant species in the chromatogram; most abundant ions in the mass sppectra.

No.	t_r_ (min)	Peak Identification	Main Ions (*m*/*z*)
1	14.9	ethylbenzene	106, **91**, 77, 65, 51
2	15.5	styrene	**104**, 89, 78, 63, 51
3	16.1	benzene, (1-methylethyl)-	120, **105**, 91, 77, 51
4	16.6	unknown	175, 146, **132**, 115, 102
5	16.7	HMDS unknown	**222**, 206, 190, 132, 74
6	17.3	HMDS unknown	**220**, 207, 188, 132, 73
7	17.4	cyclotrisiloxane, hexamethyl-	**207**, 191, 133, 96
8	17.9	cyclotetrasiloxane, octamethyl-	**281**, 265, 207, 191, 133, 73
9	18.1	benzene, 1-propenyl-	**117**, 103, 91, 77, 63, 51
10	18.2	benzene, (1-methylene-2-propenyl)-	**130**, 115, 102, 91, 77, 63, 51
11	18.5	**tetrasiloxane, decamethyl-**	295, **207**, 191, 73
12	18.6	silane, trimethylphenoxy-	166, **151**, 135, 91, 77
13	18.7	benzene, (1-methylenepropyl)-	132, **117**, 103, 91, 77, 63, 51
14	18.8	propanoic acid, 2-[(trimethylsilyl)oxy]-, trimethylsilyl ester	191, **147**, 133, 117, 73
15	18.9	acetophenone	120, **105**, 77, 51
16	19.1	acetic acid, [trimethylsilyl)oxy]-, trimethylsilyl ester	205, 177, **147**, 133, 73
17	20.0	pentanoic acid, 4-oxo, trimethylsilyl ester	173, 145, 131, **75**
18	20.1	4,6-dioxa-5-aza-2,3,7,8-tetrasilanonane-2,2,3,3,7,7,8,8-octamethyl-	294, **206**, 190, 130, 73
19	20.2	propanoic acid, 3-[(trimethylsilyl)oxy]-, trimethylsilyl ester	219, 177, **147**, 133, 116, 73
20	20.3	silane, trimethyl(4-methylphenoxy)-	180, **165**, 149, 135, 91
21	20.9	cyclopentasiloxane, decamethyl	355, 267, 251, 187, **73**
22	21.2	propanedioic acid, bis(trimethylsilyl) ester	233, 179, **147**, 73
23	21.3	pentasiloxane, dodecamethyl-	369, 353, **281**, 265, 207, 147, 43
24	21.6	unknown	281, 192, **117**, 151, 135, 115, 73
25	21.7	1-phenyl-1-(trimethylsilyloxy)ethylene	**191**, 177, 135, 103, 91, 75
26	21.9	**benzoic acid trimethylsilyl ester**	194, **179**, 135, 105, 77, 51
27	22.2	1-dimethylvinylsilyloxy-3-methylbenzene	192, **117**, 165, 151, 135, 91
28	22.5	phenylacetic acid, trimethylsilyl ester	193, 164, 91, **73**
29	22.7	**butanedioic acid, bis(trimethylsilyl) ester**	147, 172, **147**, 73
30	22.9	butanedioic acid, methyl-, bis(trimethylsilyl) ester	261, 217, **147**, 73
31	23.4	hexasiloxane, tetradecamethyl-	443, 355, 281, 267, 221, 147, **73**
32	24.1	phenylpropanoic acid, trimethylsilyl ester	222, 207, **104**, 91, 75
33	25.5	bibenzyl	182, **91**, 65
34	25.9	**1-pentene-2,4-diyldibenzen**	194, 115, **105**, 91
35	26.4	benzene, 1,1′-(1,2-dimethyl-1,2-ethanediyl)bis-	210, **105**, 91, 77
36	26.7	**benzoic acid, 4-[(trimethylsilyl)oxy]-, trimethylsilyl ester**	282, **267**, 223, 193, 73
37	27.1	benzene, 1,1′-(1,3-propanediy)bis-	196, 117, 105, **92**, 77, 65, 51
38	27.7	stilbene	**179**, 165, 152, 102, 89, 76, 51
39	27.8	**3-butene-1,3-diyldibenzene (styrene dimer)**	208, 130, 115, 104, **91**, 77, 65
40	28.0	unknown	**194**, 165, 152, 115, 91, 77, 51
41	28.4	1H-indene, 2-phenyl-	**192**, 165, 115, 91
42	28.5	1,4-benzenedicarboxylic acid, bis(trimethylsilyl) ester	310, **295**, 251, 221, 140, 103, 73
43	28.7	naphthalene, 1,2-dihydro-4-phenyl-	**206**, 191, 128, 115, 91
44	28.8	anthracene	**178**, 152, 89, 76
45	28.9	1,3-butadiene, 1,4-diphenyl-	**206**, 191, 178, 165, 128, 115, 91
46	29.4	naphthalene, 1-phenyl-	**204**, 101, 89
47	29.6	2,5-diphenyl-1,5-hexadiene	234, 143, **130**, 115, 104, 91, 77
48	29.9	pentadecanoic acid, trimethylsilyl ester	297, 145, 129, 117, **73**
49	30.4	fluoranthene, 1,2,3,10b-tetrahydro-	206, 190, **178**, 165, 152, 89, 76
50	30.5	naphthalene, 2-phenyl	**204**, 101, 89
51	30.9	hexadecanoic acid, trimethylsilyl ester	328, **313**, 145, 129, 117, 73
52	32.6	octadecanoic acid, trimethylsilyl ester	341, 145, 129, 117, **73**
53	34.6	**5-hexene-1,3,5-triyltribenzene (styrene trimer)**	312, 207, 194, 117, **91**, 77

**Table 4 polymers-13-01997-t004:** Identification of peaks in the chromatogram obtained in the Py(HMDS)–GC–MS analysis of the DCM extract of PET-4w (Figure 13). Bold: most abundant species in the chromatogram; most abundant ions in the mass sppectra.

No.	t_r_ (min)	Peak Identification	Main Ions (*m*/*z*)
1	15.3	styrene	**104**, 89, 78, 63, 51
2	16.4	unknown	175, 146, **132**, 115, 73
3	16.7	HMDS unknown	**222**, 206, 190, 132, 74
4	17.2	HMDS unknown	**220**, 204, 132, 73
5	17.3	HMDS unknown	**220**, 207, 188, 132, 73
6	17.4	unknown	223, 207, 191, **147**, 73
7	17.9	cyclotetrasiloxane, octamethyl	**281**, 265, 249, 193, 73
8	18.5	tetrasiloxane, decamethyl	295, **207**, 191, 73
9	18.6	silane, trimethylphenoxy-	166, **151**, 135, 91, 77
10	18.9	acetophenone	120, **105**, 77, 51
11	19.1	acetic acid, [trimethylsilyl)oxy]-, trimethylsilyl ester	205, 177, **147**, 133, 73
12	20.1	**2,2,3,3,7,7,8,8-octamethyl-4,6-dioxa-5-aza-2,3,7,8-tetrasilanonane**	294, **206**, 190, 73
13	21.5	unknown	**293**, 205, 146, 130, 73
14	21.8	**benzoic acid trimethylsilyl ester**	194, **179**, 135, 105, 77, 51
15	22.7	butanedioic acid, bis(trimethylsilyl) ester	247, 172, **147**, 73
16	23.4	benzoic acid, 2-methyl-, trimethylsilyl ester	208, **193**, 149, 119, 91, 65
17	23.7	naphthalene, 2-ethenyl-	**154**, 128, 76
18	24.4	unknown	442, 354, 266, 206, **146**, 130, 73
19	24.5	decanoic acid, trimethylsilyl ester	**229**, 145, 129, 117, 73
20	24.9	2-propenoic acid, 3-phenyl-,trimethylsilyl ester	220, **205**, 161, 131, 103, 75
21	25.2	unknown	275, 147, 117, **73**
22	25.6	vinyl benzoate	**105**, 77, 51
23	25.9	benzoic acid, 3-[(trimethylsilyl)oxy]-, trimethylisilyl ester	282, **267**, 223, 193, 73
24	26.0	divinyl terephthalate	**175**, 147, 132, 104, 76
25	26.7	unknown	236, **221**, 177, 147, 91
26	26.9	1,4-benzenedicarboxylic acid, methyl trimethylsilyl ester	252, **237**, 221, 163, 135, 103
27	27.3	unknown	249, **221**, 205, 170, 103
28	27.7	1,4-benzenedicarboxylic acid, ethyl trimethylsilyl ester	**251**, 221, 207, 177, 149, 103, 76
29	28.1	1,3-benzenedicarboxylic acid, bis(trimethylsilyl) ester	**295**, 279, 221, 205, 140, 103, 73
30	28.5	**1,4-benzenedicarboxylic acid, bis(trimethylsilyl) ester**	310, **295**, 251, 221, 140, 103, 73
31	29.7	unknown	265, 221, **147**, 103, 73
32	30.2	unknown	**265**, 249, 175, 149, 104
33	30.9	unknown	313, **295**, 251, 221, 149, 117, 73
34	31.3	unknown	339, **221**, 140, 103, 73
35	32.6	**2,2-bis[(4-trimethylsilyloxy)phenyl]propane**	372, **357**, 207, 73

## Data Availability

The data presented in this study are available in the Appendix A.

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
