# Peer review of "A Systematic Study on the Degradation Products Generated from Artificially Aged Microplastics"

_polymers, 2021, doi:10.3390/polym13121997_

Round 1

Reviewer 1 Report

A very informative paper about degradation products of the most relevant polymers.

I have only a few suggestions:

Page 9, line 308: "The pyrolysis profile of PS-4w and PS-0w are very similar." Why not showing both profiles, maybe in the supplementary materials part?

Page 11, line 340: "The Py-GC-MS profile of the PET-4w sample is also in this case very similar to that of the unaged polymer". Same here, why not showing both profiles?

Page 20, line 515: "the pyrolysis profiles of the extracts of PET-0w and PET-4w samples were nearly 515 identical." Also here, it would be helpful to see both profiles.

Page 23, Figure 16: the SEC chromatogram after 2 weeks (purple) is not visible. Please correct this figure.

Author Response

A very informative paper about degradation products of the most relevant polymers.

I have only a few suggestions:

Page 9, line 308: "The pyrolysis profile of PS-4w and PS-0w are very similar." Why not showing both profiles, maybe in the supplementary materials part?

We appreciated this suggestion, and in this revised version the pyrolysis profile of PS-4w is included in the Supplementary Materials (Figure S.4). Following also the suggestions by Reviewer 2, we left the pyrolysis profile of PS-0w (Figure 5) in the main text, moving Table 3 in the Supplementary Materials. All Figures and tables were renumbered when opportune.

Page 11, line 340: "The Py-GC-MS profile of the PET-4w sample is also in this case very similar to that of the unaged polymer". Same here, why not showing both profiles?

Consistently with the above comment, we added the pyrolysis profile of PET-4w in the Supplementary Materials (Figure S.7), leaving the pyrolysis profile of PET-0w in the main text.

Page 20, line 515: "the pyrolysis profiles of the extracts of PET-0w and PET-4w samples were nearly 515 identical." Also here, it would be helpful to see both profiles.

Also in this case we included the pyrolysis profile of the DCM extract of PET-0w in the Supplementary Materials (Figure S.14).

Page 23, Figure 16: the SEC chromatogram after 2 weeks (purple) is not visible. Please correct this figure.

We corrected the figure 16a, now the SEC chromatogram of the 2-week sample is more visible, even though it is very similar to the 3 weeks SEC chromatogram.

Reviewer 2 Report

Manuscript entitled “A systematic study on the degradation products generated from artificially aged microplastics” submitted by Greta Biale, Jacopo La Nasa, Marco Mattonai, Andrea Corti, Virginia Vinciguerra, Valter Castelvetro, Francesca Modugno, can be accepted for publication in Polymers Journal, after a serious major revision.

Here is a list of my specific comments:

  1. General comment 1: The utility of this study should be clearly highlighted in the manuscript.
  2. General comment 2: Pay attention on the interpretation of the experimental results. Only their presentation significantly reduces the importance of this study.
  3. Page 1, Abstract: This section is too long and should be shortened. Also, include in this section the most important experimental results to highlight the importance of this study.
  4. Page 1, Keywords: The number of keywords is too high and should be reduced.
  5. Page 1, 1. Introduction: This section should be systematized. The most important issues of this topic should be clearly described. Also, at the end of Introduction, the main objectives of this study should be detailed presented.
  6. Page 4, 3. Results and discussion: This section is too descriptive and should be systematized. The results included in this section should be properly interpreted in accordance with the objectives of this study.
  7. Page 5, line 203: “The overall procedure is schematically…”. Figure 1 should be moved in Experimental section.
  8. Page 5, line 205: “A higher rate of degradation,…”. This observation should be detailed.
  9. Page 7, line 266: “…peak assignments are listed in Table 2.”. Table 2 should be moved in Supplementary materials. Figure 4 is enough.
  10. Tables 3-5: The same observation as above.
  11. Page 26, 5. Conclusions: This section is too long and should be systematized. Delete the bullets and provide a properly description of the most important experimental results and findings included in this study.

Author Response

Manuscript entitled “A systematic study on the degradation products generated from artificially aged microplastics” submitted by Greta Biale, Jacopo La Nasa, Marco Mattonai, Andrea Corti, Virginia Vinciguerra, Valter Castelvetro, Francesca Modugno, can be accepted for publication in Polymers Journal, after a serious major revision. 

Here is a list of my specific comments:

General comment 1: The utility of this study should be clearly highlighted in the manuscript.

The introduction was improved and modified to better highlight the objectives and the innovative aspects of the described research (lines 74-87 and 103-109).

General comment 2: Pay attention on the interpretation of the experimental results. Only their presentation significantly reduces the importance of this study.

Comments especially focused on the interpretation of the obtained data and on drawing hypothesis to interpret them are present in Section 3 (lines 318-326, 349-350, 392-395, 447-451, 486-489, 601-605), and the majority of these comments have been improved and made clearer in this revised version. We systematized in a clearer manner what is data and what is interpretation, in particular modifying and integrating the comments in lines 221-226, 252-257, 264-273, 280-289, 407-410, 517-525, 620-622, 634-637, 651-652.

Page 1, Abstract: This section is too long and should be shortened. Also, include in this section the most important experimental results to highlight the importance of this study.

The abstract was significantly shortened as suggested, and a summary of the most significant results was included at the end.

Page 1, Keywords: The number of keywords is too high and should be reduced.

The number of keywords has been now reduced to six.

Page 1, 1. Introduction: This section should be systematized. The most important issues of this topic should be clearly described. Also, at the end of Introduction, the main objectives of this study should be detailed presented.

The introduction was modified to better highlight the objectives, the impact and the innovative aspects of the described research (lines 74-87 and 103-109).

Page 4, 3. Results and discussion: This section is too descriptive and should be systematized. The results included in this section should be properly interpreted in accordance with the objectives of this study.

Also to answer to a previous comment, we systematized in a clearer manner the descriptions and the comments, in particular modifying and integrating the comments in lines 221-226, 252-257, 264-273, 280-289, 407-410, 517-525, 620-622, 634-637, 651-652.

Page 5, line 203: “The overall procedure is schematically…”. Figure 1 should be moved in Experimental section.

We moved Figure 1 to Section 2.3 of Materials and Methods, where the extraction and analysis steps are described.

Page 5, line 205: “A higher rate of degradation,…”. This observation should be detailed.

The comment was rearranged and made clearer, as follows (lines 212-219):

The extraction yields increased steadily throughout the investigated aging time for LDPE, PP, and PS. The higher sensitivity of these polymers towards aging can be relat-ed to the presence of tertiary and benzyl carbon atoms, which are more prone to be at-tacked by free radical species directly or indirectly generated by photo-irradiation, and therefore more susceptible to undergo C-C bond cleavage as a result of secondary pro-cesses (e.g. β-cleavage of oxy-radicals generated upon decomposition of per-oxy-radicals, the latter resulting from oxygen pickup by the primary radicals produced by H-abstraction)”.

Page 7, line 266: “…peak assignments are listed in Table 2.”. Table 2 should be moved in Supplementary materials. Figure 4 is enough.

Tables 3-5: The same observation as above.

We moved Tables 2-5 in the Supplementary Materials (Tables S.1-S.4). Table numbering was rearranged accordingly.

Page 26, 5. Conclusions: This section is too long and should be systematized. Delete the bullets and provide a properly description of the most important experimental results and findings included in this study.

The conclusions section was completely revised, shortened, and rewritten according to the given suggestions.

Round 2

Reviewer 2 Report

Manuscript entitled “A systematic study on the degradation products generated from artificially aged microplastics” submitted by Greta Biale, Jacopo La Nasa, Marco Mattonai, Andrea Corti, Virginia Vinciguerra, Valter Castelvetro, Francesca Modugno, can be accepted for publication in Polymers Journal, after a minor revision.

Here is a list of my specific comments:

  1. Page 6, line 222: “This can be interpreted as a result of some radical scavenging…”. This paragraph should be reworded.
  2. Page 8, line 289: “In order to detect the presence of such oxidized…”. Where are the results obtained under these conditions???
  3. Page 11, line 395: “Even though the general features…”. Provide an explanation for this.
  4. Page 23, 5. Conclusions: This section is still too long and should be systematized. Also, the most important experimental results should be included in this section to highlight the importance of this study.

Author Response

Page 6, line 222: “This can be interpreted as a result of some radical scavenging…”. This paragraph should be reworded.
The sentence was rewritten as follows:

“This result can be interpreted as an effect of possible radical scavenging associated with the presence of the monosubstituted phenyl ring and/or with the higher glass transition temperature (Tg) of PS: a lower diffusional mobility of free radical species and thus an induction time associated with slower increase of their concentration is typically associated with free radical transfer and oxygen pickup.”

Page 8, line 289: “In order to detect the presence of such oxidized…”. Where are the results obtained under these conditions???

The results of the soluble fraction are reported in the Section 3.1 of the manuscript. To avoid confusion in the reader the sentence was removed.

Page 11, line 395: “Even though the general features…”. Provide an explanation for this.

The following explanation was added in the relative section:

…probably due to the different intermolecular forces in the structures of the two polymers and consequence differences in the ageing behaviors.”

Page 23, 5. Conclusions: This section is still too long and should be systematized. Also, the most important experimental results should be included in this section to highlight the importance of this study.

The conclusions were shortened to half of their original length in the first round of revisions, and now they consist of 445 words only. They are divided in subparagraphs for type of polymers, containing a very short summary of the relevant results obtained. We further revised them; however it was not possible to cut more than this, and we do not think that the paper will benefit of a further modification of this section.